# The Effect of a Brief Physician-Delivered Neurobiologically Oriented, Cognitive Behavioural Therapy (Brief-CBT) Intervention on Chronic Pain Acceptance in Youth with Chronic Pain—A Randomized Controlled Trial

**DOI:** 10.3390/children9091293

**Published:** 2022-08-26

**Authors:** Mark K. Simmonds, Bruce D. Dick, Michelle J. Verrier, Kathy L. Reid, Kathryn Jamieson-Lega, Keith J. Balisky, Alison E. Davey, Katherine M. Freeman

**Affiliations:** 1Department of Anesthesiology and Pain Medicine, 2-150 Clinical Sciences Building, University of Alberta, Edmonton, AB T6G 2G3, Canada; 2Department of Anaesthesiology and Pain Medicine, University of Alberta, Edmonton, AB T6G 2R3, Canada; 3CBI Health–Edmonton Southwest, 6103 Currents Drive NW, Suite 310, Edmonton, AB T6W 2Z4, Canada; 4Sarum Ward, Salisbury District Hospital, Odstock Rd, Salisbury SP2 8BJ, UK; 5Department of Psychological Sciences, School of Psychology, University of East London, Stratford Campus, London E15 4LZ, UK

**Keywords:** adolescent, chronic pain, anxiety, compassion, cognitive therapy

## Abstract

At the Stollery Children’s Chronic Pain Clinic, new referrals are assessed by an interdisciplinary team. The final part of the intake assessment typically involves an explanation and compassionate validation of the etiology of chronic pain and an invitation to the youth to attend the group outpatient Cognitive Behavioural Therapy (CBT) program, called Pain 101, or to individual outpatient CBT. It was hypothesized that a brief physician-delivered CBT (brief-CBT) intervention at first point of contact improves subsequent pain acceptance. Using a randomized double blinded methodology, 26 participants received a standard intake assessment and 26 the standard assessment plus the brief-CBT intervention. Measures were taken at three points: pre and post-intake assessment and after Pain 101 or individual CBT (or day 30 post-assessment for those attending neither). The primary outcome measure was the Chronic Pain Acceptance Questionnaire—Adolescent version (CPAQ-A). Comparing pre and post-intake measures, there was a significant (*p* = 0.002) increase in the CPAQ-A scores—four-fold more in the brief-CBT intervention group (*p* = 0.045). Anxiety (RCADS-T Score) was significantly reduced post-intake and significantly more reduced in the intervention group compared to the control group (*p* = 0.024). CPAQ-A scores were significantly increased (*p* < 0.001) (N = 28) and anxiety (RCADs-T) was significantly reduced by the end of Pain 101 (*p* < 0.003) (N = 29) as was fear of pain as measured by the Tampa Scale for Kinesiophobia (*p* = 0.021). A physician-delivered brief-CBT intervention significantly and meaningfully increased CPAQ-A scores and reduced anxiety in youth with chronic pain. Furthermore, CBT through Pain 101 is effective at increasing acceptance, as well as reducing anxiety and fear of movement.

## 1. Introduction

When a young person suffering chronic pain first comes into contact with a physician, there is a critical yet brief opportunity for the physician to empathetically validate the youth’s pain experience, help settle their anxiety, and build an early bridge to effective biopsychosocial chronic pain management strategies [1,2,3,4,5]. Successful validation depends upon a number of factors: (i) a trusting therapeutic alliance with the health care providers (ii) the youth being taken seriously and given time and latitude to explain their symptoms, (iii) the youth being listened to and believed that their pain experience is genuine and (iv) the youth receiving a clear and plausible medical explanation of the cause of their pain to enhance their own understanding of their pain [6,7,8,9,10,11,12]. Acceptance and adherence with any subsequent treatment offered to a youth and their family could be potentially influenced by this validation and explanation. In a multidisciplinary pain team, the physician is ideally placed to provide the biomedical explanation as youth and their parents often look to them for expert medical advice, reassurance, and medical closure [13,14]. Additionally, in Western society, physician opinion and recommendations have the potential to be of particular significance for youth and their families [15]. It has been previously highlighted how children with chronic pain often experience negative, judgmental and stressful encounters with clinicians, related to clinicians’ struggles to secure a diagnosis and confidently explain chronic pain; exposing a need to recruit clinicians, including physicians, to remedy this issue [7]. This rationale provided the impetus for our study.

Accumulating scientific evidence implicates the central nervous system in the etiology of chronic pain and, in particular, increased nervous system sensitivity appears to be a key biomedical component [16,17]. It is not known how well-educated patients are regarding this information, and it is possible that dissemination of this information may dispel some of the myths regarding the etiology of chronic pain (e.g., that patients are imagining their symptoms or that the cause is psychosomatic) [18,19]. A previous study looking at the efficacy of brief education intervention found that neurophysiology education led to some normalization of attitudes and beliefs about pain, a reduction in catastrophizing, and improvement in physical performance [20]. However, this study was on adult participants. Emerging brief psychological interventions for pediatric chronic pain have been reported as being feasible, acceptable and effective in improving functioning and psychological well-being and there are also small number of promising pain neuroscience educational programs emerging for children with chronic pain but there remains a need to explore further the efficacy of such modalities and expand their utility, if appropriate [21,22,23,24].

Physicians are in a strong position to validate a youth’s pain experience, explain to a youth and their caregivers the etiology of their chronic pain in neurobiological terms and, without delay, begin the process of active chronic pain management through the use of a brief Cognitive Behavioural Therapy (brief-CBT) intervention. Physicians play a key role in health care including chronic pain management and hold a position that has the potential to garner and augment treatment participation in youth and their families in tandem with other pain team members, including psychologists. CBT is one potential keystone of pain management in youth and successful outcomes are now being realized [25,26]. CBT has been used with benefit in managing chronic pain through cognitive restructuring, behavioural management and is well described in the literature [27]. Our CBT program focused on key areas targeting education, reducing physiological arousal, thought management, mood management, stress management, and attention regulation through mindful awareness.

At our pediatric pain clinic, all new referrals are comprehensively assessed by an interdisciplinary team in a single sitting over approximately 90–120 min, which, is our standard intake consultation. The core members of the team include the pain physician, physiotherapist, nurse–practitioner, and psychologist. During the final part of the consultation the pain physician and other team members offer diagnoses where possible and, through full informal discussion, give a verbal explanation of the etiology and biology of chronic pain to the youth and caregivers. Additionally, a pain management plan is formulated and discussed with the youth and caregivers and ample time is allotted to answer all questions. The plan typically involves inviting the youth to attend the group CBT chronic pain management program (called Pain 101) over the ensuing weeks or to attend individual outpatient CBT sessions. Pain 101 is a ten-session, psychology-led, children’s outpatient program. Youth who are actively suicidal are referred immediately for individual treatment prior to being invited to Pain 101. Youth who are primarily interested in pursuing biomedical intervention for pain management are informed that they are welcome to partake in a future group series if they so desire.

Our research study explores the hypothesis that a dedicated, physician-delivered, neurobiologically based, brief-CBT intervention improves chronic pain acceptance in youth experiencing chronic pain. Acceptance has been defined as a positive adaption to experiencing pain without taking actions to control it, and persisting with activity, openly in the presence of pain which are captured in the two subscales of the Chronic Pain Acceptance Questionnaire (CPAQ), which was the study primary outcome [28]. It has been demonstrated that higher levels of acceptance in adolescents is associated with lower levels of disability and distress [28]. Our study intervention was provided in the form of a formal standardized interactive PowerPoint^®^ presentation to the youth and caregivers and was given immediately following the initial interdisciplinary team assessment. This program involved pain neuroscience teaching as well as a discussion of how cognitive-behavioural principles are applicable to chronic pain self-management. To the best of our knowledge this is the first randomized controlled trial looking at the therapeutic outcomes from a brief, physician delivered neuroscience-based CBT intervention in youth with chronic pain.

## 2. Materials and Methods

### 2.1. Ethical Approval

This study was approved by the University of Alberta Health Research Ethics Board. (Ethics information descriptor Pro00015844).

### 2.2. Consent and Funding

Informed consent was obtained from all participants in accordance with the University of Alberta’s Health Research Ethics Board requirements. Funding was provided by the University of Alberta department of Anesthesiology and Pain Medicine.

### 2.3. Participants

Inclusion criteria were age 10–18 years with a history of chronic pain of greater than 3 months, which is in line with the validated age range of the CPAQ-A questionnaire and also the age criterion for admission to Pain 101 [28]. Individuals with a medical history that included significant head injury, neurological disorder, or disease known to significantly impair cognitive functioning were excluded if the level of impairment was such that taking part in study requirements was significantly hindered or impossible. Young people who attend our clinic have diverse sources of chronic pain and are all treated along similar lines. All types and locations of chronic pain were considered eligible for this study. The study was limited to the English language and study information was written with a Flesch Kincaid reading level equal to Grade 8 or less.

A total of 113 participants from individuals referred to the Stollery Children’s Hospital Pediatric Chronic Pain Clinic were invited to enroll in the research study. Out of this sample, 16 declined to participate and 5 did not meet the inclusion criteria to enroll. Out of the 92 randomized participants, 26 of the 44 who were allocated to the intervention completed all of the study requirements, and 26 of the 48 who were not selected to view the brief-CBT presentation completed all study requirements. Consecutive participants were recruited from 1 December 2010 to 21 August 2013. Full details on enrollment are included in the CONSORT flow diagram.

### 2.4. Sample Size Determination

Allowing for a large size effect, alpha = 0.05, and power of 0.80 a sample size of 52 was required for our study [29]. Two groups of 26 were compared; one group receiving the formal physician-lead brief-CBT intervention and the other group receiving standard care without the intervention.

### 2.5. Intervention

The intervention was in the form of a narrated interactive visual presentation using Microsoft PowerPoint^®^ slides. The slideshow was divided into three sections (1) pain pathway anatomy, (2) a neurobiological explanation of chronic pain and (3) a brief description of the components of CBT for pain management. The key features for an effective presentation were included as recommended by Hoffmann and Cutilli [30,31]. The intervention took approximately 10–15 min to deliver.

Following the standard interdisciplinary intake consultation, randomized participants were asked to enter a room where the chronic pain physician presented the visual and interactive presentation, allotting 5 min for questions at the end. A script was used during the presentation to maintain consistency of information between participants. The efficacy of the intervention was the main focus of our study.

### 2.6. Measures

Measures were chosen as recommended in the international expert-led PedIMMPACT recommendations for core outcomes and domains in pediatric chronic pain clinical trials [32].

Demographics and Pain History

Details including the youth’s age, sex, years of education, pain location, chronicity of pain (date of pain onset) were collected.

Visual Analogue Scale (VAS)

The youth was asked to rank their current pain by drawing a vertical line on a 100 mm horizontal line using the endpoints of “no pain” and “worst pain imaginable” [33].

Chronic Pain Acceptance Questionnaire—Adolescent version (CPAQ-A) [28]

The CPAQ-A questionnaire has been designed to measure acceptance of pain and has been found to reliably predict pain-related disability and distress. This self-report questionnaire is validated for age 10–18 years. Respondents were required to report how true they believe each of the 20 statements represent their thoughts, using 0–4 scale where 0 corresponds to “Never true” and 4 “Always true”. CPAQ-A measures two components: persisting with activity in the presence of pain (activity engagement) and experiencing pain without attempting to control it (pain willingness).

Functional Disability Inventory (FDI) [34]

This scale consists of 15-items where the child and parent(s) are asked to report the youth’s level of physical trouble or difficulty when performing various everyday activities. Responses to each item are scored from 0–4, where 0 means “No trouble” and 4 means “Impossible”.

Revised Children’s Anxiety and Depression Scale (RCADS) [35,36]

This is a 47-item self-report questionnaire with scales relating to separation anxiety, social phobia, generalized anxiety disorder, panic disorder, obsessive compulsive disorder, and major depressive disorder. Respondents are asked to rate how often each item applies to them. Items range from 0–3, where 0 means “never”, 1 “sometimes”, 2 “often” and 3 “always”. A number of investigators have illustrated support for the RCADS in non-referred samples of youth [35].

Tampa Scale of Kinesiophobia—Child version (TSK) [37,38]

This scale has been adapted for use in children and measures fear of movement and re-injury in individuals with chronic pain. The adult version of this scale has been widely validated and this child version holds considerable promise.

Acceptability Questionnaire

This is a manipulation check completed by the participants and inquires about issues such as ease of reading, length, clarity, informative content, and believability [39]. This was used to test the practicality and informative nature of our visual CBT intervention. A similar questionnaire was used on the standard care group.

Satisfaction Survey

Parents/caregivers were asked to rate on a Numerical Rating Scale (i) satisfaction of the explanation given for their youth’s pain, (ii) how believable this information was and (iii) how strongly they felt further diagnostic tests were needed. The rationale for using this questionnaire is that caregivers who are in pursuit of further biomedical diagnostics are less likely to be accepting of CBT.

### 2.7. Procedure

Youth who were being seen at the Stollery Children’s Hospital Pediatric Chronic Pain Clinic and their parents/caregivers were approached to take part in the study. A research assistant was responsible for recruiting and randomizing participants. Each youth and parent were asked to read an information sheet describing the study procedures. The opportunity to discuss the information and ask questions was provided, and questions or concerns were addressed by the investigator. Prior to enrollment, each youth and parent were required to sign the study consent and assent forms indicating that they understood the study procedures.

Youth who chose to participate in the study were randomized in one of two groups: one group received the standard interdisciplinary team intake assessment, (in a form that has been practiced in our clinic for several years) and the other group received the standard intake assessment plus the brief-CBT intervention. All individuals involved in this study were blinded to randomization allocation until the end of standard treatment except the research assistant who carried out the randomization procedure. All study participants were required to fill out the questionnaire package, immediately before the intake assessment, after having read the information letter and signed the consent and assent forms. Initial data gathered from these questionnaires was used as a baseline for follow-up comparison. Those referred to Pain 101 repeated the battery of questionnaires on day 1 before commencing Pain 101 and after the final Pain 101 session. Likewise, those referred for individual CBT sessions repeated the questionnaires on day 1 and after completion of their sessions. Those who were neither referred to Pain 101 nor to individual CBT repeated the questionnaires just once more on day 30 after their intake assessment. For convenience, the first and second battery of questionnaires are referred to as pre and post-intake assessment questionnaires, respectively. The questionnaires were returned to study personnel in a self-addressed, stamped envelope.

A debriefing form was also provided to all participants. An opportunity for the non-intervention group to receive the brief-CBT intervention was scheduled at a later date when desired by participants.

## 3. Results

### 3.1. Statistical Analysis

Analyses were completed using IBM SPSS Statistics software (2015). Within- and between-group changes in fear of movement, pain, acceptance, and pain-related disability was examined using repeated measures ANOVAs. All post hoc testing that examines the underlying effects of individual factors was conducted using Bonferroni corrections to control for the inflation of alpha. Stepwise logistic regressions were also conducted to ascertain the strongest baseline predictors of post-intervention pain acceptance, fear of movement, and anxiety.

### 3.2. Demographics

See Table 1. There were 52 participants who completed the full study (12 males and 40 females). A total of 44 out of 52 participants experienced daily pain, 7 experienced pain between 1 and 5 times a week and 1 subject reported their pain recently settling. The participants’ chronic pain was classified according to bodily location. See Table 2. No significant demographic differences were found between the intervention and control group. Young people who enrolled in Pain 101 missed significantly less school than youth who did not enroll in Pain 101 (*p* = 0.023) and no significant difference in mood (depression or anxiety) existed between these two groups.

There were 26 participants in each of the intervention and control groups. A total of 29 participants out of the 52 went on to attend Pain 101, 8 received individual CBT and 15 no CBT. The baseline measures are reported in Table 3.

### 3.3. Acceptability and Satisfaction Questionnaires

Of the participants who were given the brief-CBT presentation, >90% affirmed that they found the presentation interesting, clear, easy to read, the right length, likeable and understood the role of the nervous system in chronic pain and where pain is felt. Additionally, >80% affirmed that they believed most of what was said and would recommend the presentation to others. See Figure 1. The satisfaction questionnaire suggests high levels of satisfaction and believability in the explanation for the child’s chronic pain and low scores for belief in the need for further medical tests in both the intervention and control groups. See Table 4.

### 3.4. CPAQ-A

See Table 5. Comparing pre- and post-intake assessment scores there was a significant (*p* = 0.002) and meaningful increase in the CPAQ-A Total scores but more than four-fold in the brief-CBT intervention group (*p* = 0.045). CPAQ-A Total scores further increased significantly by the end of Pain 101 (*p* < 0.001). CPAQ-A Activity Engagement increased significantly across groups (*p* = 0.023) but the intervention group did not improve significantly more than the control group (*p* = 0.473). CPAQ-A Activity Engagement scores significantly increased overall following Pain 101 (*p* = 0.001). CPAQ-A Willingness significantly improved across groups (*p* = 0.038) and this improvement was greater in the intervention group (*p* = 0.021). There was only a statistical trend toward an increase in CPAQ-A Willingness following Pain 101 (*p* = 0.061). See Table 6.

### 3.5. Anxiety

Comparing pre- and post-intake assessment scores there were very large and statistically significant (*p* < 0.000) drops in anxiety (RCADS T-Scores) with a greater decrease in the intervention group (*p* = 0.024) compared to the control group. See Table 7. Anxiety was reduced significantly by the end of Pain 101 (*p* < 0.003). See Table 8.

### 3.6. Functional Disability, Kinesiophobia and Depression

Comparing pre and post-intake assessment scores, there was no significant reduction in FDI scores both child-rated and parent-rated. However, comparing pre- and post-intake assessment scores, fear of movement as measured by the TSK was reduced (*p* = 0.021) and further reduced following Pain 101 (*p* = 0.018). See Table 9 and Table 10. Depression (RCADS) was not significantly reduced following Pain 101 and this is consistent with previous aggregate data from our Pain 101 CBT groups.

### 3.7. Regression Analyses

Stepwise linear regressions aimed at exploring baseline (pre-intake assessment) factors that predicted post-intervention (brief-CBT presentation) levels of acceptance and fear of movement found two key effects. Baseline levels of children’s ratings of their functional disability significantly predicted (*p* = 0.000) post-intervention levels of acceptance (CPAQ-A Total) such that children with higher levels of disability reported lower levels of acceptance. That factor accounted for 31.1% of the variance in the regression. As well, baseline ratings of fear of movement significantly predicted post-intervention acceptance (*p* = 0.021) where higher baseline fear of movement predicted lower acceptance. Kinesiophobia accounted for 7.9% of variance in that regression. See Table 11 for details. Furthermore, post-presentation fear of movement was found to be significantly predicted by baseline levels of acceptance (*p* = 0.000) accounting for 27.5% of variance. Youth reporting less acceptance at baseline reported higher fear of movement after the intervention. Additionally, baseline levels of depression significantly predicted fear of movement later (*p* = 0.022) and accounted for 7.6% of variance. Young people with higher levels of depression at baseline reported higher fear of movement later. See Table 12 for details.

## 4. Discussion

The motives for this study arose firstly from our clinical observation of the recurring narratives that youth presented during the initial consultation of our intake clinics; in particular their accounts of how their chronic pain was either disbelieved or discredited by their treating physicians. We noticed that many young people arrived without an adequate biomedical explanation or diagnosis of their chronic pain. Although the majority were seeking guidance regarding non-medical methods of managing their pain, few had an adequate understanding of the biopsychosocial model underpinning their pain, disability and suffering and the psychobehavioural management strategies that are currently recommended. Many arrived with significant emotional distress and sometimes hopelessness and expressed the view that our clinic was their final hope after experiencing a long line of disappointing medical consultations, physical investigations and treatments. We noticed that, on the whole, youth and their families were truly motivated to change in order to regain function and improve quality of life. Our belief is that our observations and experience in this regard are not unique.

Furthermore, as a team we were paying attention in the initial years of our clinic to clearly identify and define the roles of our team members. The roles of the physician to review the existent medical information, take a fresh medical history and carry out a focused physical examination were seen as unquestionably essential; but the issue remained of how best to transition children from seeking a pure biomedical solution towards commitment to an effective psychobehavioral pain management program. This study highlighted the considerable potential of a simple and brief CBT-based intervention shared by a physician. There are several potential implications of our findings including the importance of physicians being involved in health education, including pain education. Further, this study sheds some light on the benefits of physician education as an intervention early in the team treatment process. This study aimed to address the question of how a physician could effectively facilitate the enrolment of young people into a pain management program and affirmed that a physician-delivered brief-CBT intervention studied yields positive outcomes.

Our brief-CBT presentation information was not based on any previously published studies providing brief education intervention for chronic pain. The presentation content was based upon the scientific literature and expert opinion regarding the etiology of chronic pain. It was based on information available in other well described materials including “Explain Pain” [40]. We accept that this subjectivity may be perceived as a drawback.

We questioned ourselves as to why we chose to name our intervention brief-CBT and not psycho-education. We debated this but concluded that our intervention specifically targeted the common negative core belief ‘my pain is not real’ by exploring the neurobiology of chronic pain and through validation allowing the child and family an opportunity to become open to the possibility of positive behavioural changes through our formal CBT program [41]. We see the recent growth in pain neuroscience education for youth with chronic pain as a relevant and important movement [39].

Acceptance of pain is a predictor of pain-related disability and treatment outcome in adolescents with pain, and CPAQ-A has been used in clinical studies to measure therapeutic efficacy [42,43]. In our study, there was an impressively significant increase in CPAQ-A Total scores and reduction in anxiety comparing baseline to post-assessment scores; more-so in the intervention group, which, was a finding consistent with our subjective view of the emotional relief expressed by many of the children in our assessment clinic and their willingness to commit to the CBT programs on offer. There were further gains in anxiety reduction and increases in CPAQ-A Total scores following Pain 101, which makes clinical sense and is also consistent with our experience and existent scientific literature, which reports on the efficacy of CBT in reducing anxiety in children with chronic pain [26]. We would not necessarily expect significant changes in FDI with a brief-CBT intervention such as ours and this was apparent in our data, however, the increase in acceptance scores would be consistent with a prediction of an improvement in later pain-related disability over the course of longer-term treatment.

There were significant improvements in kinesiophobia following the assessment process and following Pain 101, and although this study did not find a reduction in functional disability scores following Pain 101, evidence does point to an association between kinesiophobia and the development of disability over time; at least in adulthood [44]. There was no significant improvement in depression scores following Pain 101 and this is consistent with our experience and current research, which finds that depression in children and adolescents with chronic pain is difficult to successfully treat with psychological therapies alone over a relatively limited time period [26]. Given the relationship between acceptance and disability, the findings of the regression analysis predicting lower post-intervention acceptance with higher baseline disability and fear of movement was not surprising. Further, the unsurprising finding that mood, fear of movement, and acceptance were all associated, points to the complex interactions of these factors.

As a team we have received strong feedback that young people and their families experience huge relief from anxiety and doubt when the physician spends dedicated time validating the youth’s pain experience as being a genuine biomedical problem and explaining the cause of their pain, particularly when there is a focus on exploring validation with the youth’s family. Although this was not measured in our study it is feasible that youth and their families are, as a consequence, more likely to enroll in psychobehavioural programs to manage their chronic pain. This is an important consideration in the design of future research.

Several questions remain. There is an enormous literature on the scientific underpinnings of CBT and its effect on human performance including thoughts, mood, and behaviour. Much remains to be learned regarding the neurobiological underpinnings and cognitive processes involved in CBT in this and thousands of other studies. Was the positive effect we found in this study more to do with the psychoeducational material itself or simply the result of dedicated therapeutic time with the pain physician or a potentially powerful combination of the two? The difference of receiving physician directed treatment may have added to a placebo response and should be studied in future research. Would there be a similar positive effect if the brief-CBT intervention was delivered by a non-physician? Would the clinical effect of the intervention be amplified by further involvement of the pain physician in the opening and subsequent sessions of the formal CBT course Pain 101? This study is supportive of the assertion that interested physicians, who, are willing to dedicate time offering psychoeducational or brief-CBT interventions to youth with chronic pain are well positioned to facilitate positive outcomes and pave the way for youth to engage in psychological therapies for chronic pain management.

## 5. Conclusions

A 10-to-15 min physician-delivered brief-CBT intervention at the intake assessment significantly and meaningfully increases CPAQ-A scores and reduces anxiety in youth with chronic pain. Cognitive Behavioural Therapy through our formal pain management program Pain 101 is effective at further reducing anxiety and fear of movement including in young people with chronic pain in diverse body locations.

## Figures and Tables

**Figure 1 children-09-01293-f001:**
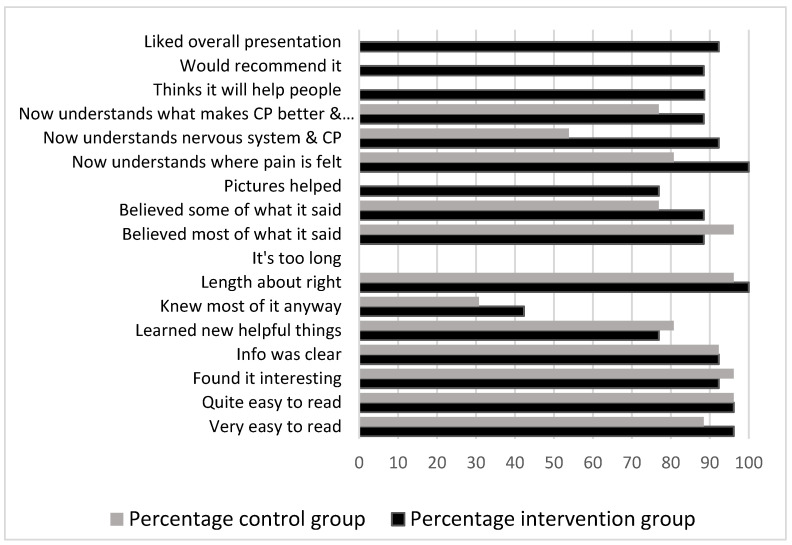
Completed Acceptability Questionnaires. Percentage responses in the intervention and control groups.

**Table 1 children-09-01293-t001:** Subject demographic characteristics.

	N	Mean	Std. Deviation
Age	52	15.00	1.66
School Grade	52	9.85	1.53
Pain Duration (years)	52	4.35	4.57
Pain VAS 0-10 (initial assessment)	52	5.19	2.53
Pain Intensity VAS 0-10 (bad day)	52	8.50	0.98
Pain Intensity VAS 0-10 (good day)	52	4.00	1.97
Pain Intensity VAS 0-10 (average)	52	5.66	1.91
School Absence (days per week)	44	1.19	1.64
School Absence (days total this year)	45	33.48	43.25

**Table 2 children-09-01293-t002:** Bodily location of pain.

Bodily Location of Pain	Number of Subjects (N)
Abdominal	2
Head	10
Abdomen and Head	1
Head and joint/spine	6
Jaw	1
Joint and Spine	20
Widespread	6
Mixed-Joint, Spine, Abdomen, Head, Chest	6

**Table 3 children-09-01293-t003:** Baseline psychological measures.

	All Subjects in Final Analysis			Enrolled to Pain 101			Not Enrolled to Pain 101		
	N	Mean	Std. Deviation	N	Mean	Std. Deviation	N	Mean	Std. Deviation
RCADS T-Score depression	52	59.62	14.57	29	58.59	14.87	23	60.91	14.41
RCADS T-Score total anxiety	52	50.11	11.05	29	51.28	10.48	23	48.65	11.80
RCADS T-Score total anxiety and depression	52	52.58	11.54	29	53.24	11.15	23	51.74	12.21
FDI child form total	52	18.77	12.30	29	18.31	12.89	23	19.34	11.77
FDI parent form total	52	20.63	13.59	29	21.72	13.96	23	19.26	13.29
CPAQ-A total score	52	37.31	12.96	29	39.34	12.63	23	34.74	13.20
TSK total score	47	28.19	5.22	26	27.81	5.83	21	28.67	4.43

**Table 4 children-09-01293-t004:** Parent/Caregiver satisfaction survey Numerical Rating Scores.

	Brief-CBT Intervention N = 26		Control Group N = 26	
	Mean	Std. Deviation	Mean	Std. Deviation
How satisfied were you today with the explanation of the cause of your child’s pain? 0 = not satisfied to 10 = very satisfied.	8.50	1.24	8.73	1.56
Do you believe that your child needs more medical tests to make a diagnosis of their chronic pain? 0 = Strongly believe “No” to 10 = strongly believe “Yes”.	3.44	3.44	3.35	2.97
How strongly do you believe the explanation the team gave for your child’s chronic pain? 0 = Do not believe explanation at all to 10 = very strongly believe.	8.85	1.28	8.58	1.72

**Table 5 children-09-01293-t005:** CPAQ-A scores pre and post-intake assessment.

	1 = Brief-CBT Intervention 2 = Control Group	Mean	Std. Deviation	N
CPAQ-A—Total	1	36.85	13.09	26
Pre-intake assessment	2	37.76	13.08	26
	Total	37.30	12.96	52
CPAQ-A—Total	1	42.19 **	13.67	26
Post-intake assessment	2	38.98 **	13.18	26
	Total	40.59	13.39	52

Comparing pre- to post-intake assessment CPAQ-A scores, there was a significant increase in both groups ** (*p* = 0.002) but with more than a four-fold increase seen in the brief-CBT intervention group (*p* = 0.045).

**Table 6 children-09-01293-t006:** CPAQ-A scores in those receiving Pain 101 ^a^.

	Mean	Std. Deviation	N
CPAQ-A—Total. Pre-intake assessment	39.00	12.72	28
CPAQ-A—Total. CBT baseline assessment	40.95	12.71	28
CPAQ-A—Total. Post Pain 101	45.86 **	13.68	28

^a^ Cognitive Behavioural Therapy. CPAQ-A—Total is increased significantly by the end of Pain 101 ** (*p* = 0.001).

**Table 7 children-09-01293-t007:** Anxiety (RCADS T-Score); pre and post- intake assessment.

	1 = Brief-CBT Intervention 2 = Control Group	Mean	Std. Deviation	N
RCADS Total Anxiety T-Score pre-intake assessment	1	48.96	9.27	25
	2	51.31	12.79	26
	Total	50.16	11.16	51
RCADS Total Anxiety T-Score post-intake assessment	1	25.04 **	12.53	25
	2	33.46 **	20.94	26
	Total	29.33	17.68	51

Comparing pre and post-intake assessment scores in both groups, large statistically significant decreases in anxiety (RCADS T-Scores) were seen ** (*p* < 0.000) but greater in the brief-CBT intervention group compared to the control group (*p* = 0.024).

**Table 8 children-09-01293-t008:** Anxiety (RCADS T-Scores) in those receiving Pain 101.

	Mean	Std. Deviation	N
RCADS Total Anxiety T-Score pre-intake assessment	51.28	10.48	29
RCADS Total Anxiety T-Score CBT baseline assessment	50.17	10.56	29
RCADS Total Anxiety T-Score following Pain 101	47.07 **	11.07	29

Anxiety is reduced significantly by the end of Pain 101 ** (*p* < 0.003).

**Table 9 children-09-01293-t009:** Fear of movement, as measured by the Tampa Scale for Kinesiophobia (TSK) pre and post- intake assessment.

	1 = Brief-CBT Intervention 2 = Control Group	Mean	Std. Deviation	N
TSK pre-intake assessment	1	28.18	5.30	22
	2	28.33	5.32	24
	Total	28.26	5.25	46
TSK post-intake assessment	1	27.30	5.00	22
	2	27.27	4.50	24
	Total	27.28	4.69	46

Fear of pain as measured by the TSK was reduced across both groups (*p* = 0.021).

**Table 10 children-09-01293-t010:** Fear of movement, as measured by the Tampa Scale for Kinesiophobia (TSK) in those receiving Pain 101.

	Mean	Std. Deviation	N
TSK pre-intake assessment	27.81	5.83	26
TSK baseline assessment	27.10	4.96	26
TSK following Pain 101	25.25 *	6.43	26

Fear of movement is significantly reduced following Pain 101 * (*p* = 0.018).

**Table 11 children-09-01293-t011:** Regression values for predictors of CPAQ-A Total at post-intake assessment.

Measure	B	SE B	Beta	t Score	Significance (*p*)
CPAQ-A Total (pre-intake assessment)	−0.125	0.048	−0.357	−2.626	0.012
RCADS-Depression (pre-intake assessment)	0.241	0.102	0.322	2.371	0.022

**Table 12 children-09-01293-t012:** Regression values for predictors of fear of movement (TSK), at post-intake assessment.

Measure	B	SE B	Beta	t Score	Significance (*p*)
FDI ^a^—Child Form—Total Score (pre-intake assessment)	−0.521	0.138	−0.468	−3.782	0.000
Tampa Scale for Kinesiophobia—Total Score (pre-intake assessment)	−0.788	0.330	−0.295	−2.387	0.021

^a^ Functional Disability Inventory.

## Data Availability

Data is available through the authors per University of Alberta Health Research Ethics Board guidelines.

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
