# Peer review of "The Effect of a Brief Physician-Delivered Neurobiologically Oriented, Cognitive Behavioural Therapy (Brief-CBT) Intervention on Chronic Pain Acceptance in Youth with Chronic Pain—A Randomized Controlled Trial"

_children, 2022, doi:10.3390/children9091293_

Round 1
Reviewer 1 Report
Thank you for the opportunity to review the manuscript The effect of a brief physician-delivered neurobiologically-oriented, Cognitive Behavioral Therapy (Brief-CBT) intervention on chronic pain acceptance in youth with chronic pain-a randomized control trial. This is a well written manuscript and was mostly easy to follow. There is value in underscoring the importance of validation and explanation of chronic pain in its therapeutic impact on outcomes post interventions offered.
It would be important to clearly state in the intro that this is the first study (if that is the case) that looks at brief pain neuroscience education plus intro to CBT intervention study in assessing outcomes. Why acceptance leads to better outcomes can be flushed out a bit more in the intro to clarify the aims/hypotheses of this study. Overall, I believe this manuscript will move our field forward and should be published with the following suggestions for improvement/edits.
Introduction:
Consider referring to caregivers vs carers
How did the study account for placebo effects? Looks like it was just the intervention group that was getting something “extra” which could trigger a strong placebo impact. If both groups got some kind of intervention one receiving a sham intervention and the other the brief CBT intervention, we could feel more comfortable in the results of this randomized study. I see you discussed this briefly in the discussion, section without reference to placebo effect.
I am curious if after the intervention, more of those patients signed up for Pain 101 vs those that didn’t get the intervention. That would be super interesting to look at and a meaningful contribution to our knowledge base. Was the rate of adherence to recommendations/completion of Pain 101 explored?
Procedure:
Would be helpful to clarify that both individual treatment and group CBT are referred to as PAIN 101. Also, are all patients recommended Pain 101. What determines their attendance? Did the patients who were recommended attend/complete the 10 sessions. Was there a difference in those that completed vs not? Do all patients get Pain 101?
Results:
“Young people who did not engage in psychological treatment missed significantly less school than youth who enrolled in Pain 101 or who engaged in individual psychology treatment sessions (p=.023).” This sentence is confusing-I suggest removing the double negative.
“No significant difference in mood (depression or anxiety) existed between these subgroups”: which subgroups are being compared here. Those that received the intervention vs not? Please clarify
Figure 1: the “yes” on the Y axis is confusing and I am not sure needed. Labeling that axis as % would suffice. Have a keep for what the yellow/black groups are would be helpful.
Where are the rates of “its too long”
So, the group that didn’t get the intervention, believed most of what was said and learned more vs those that got the intervention? This needs to be clarified and potentially explained if there was a significant difference.
Were any of these differences between groups statistically significant?
Table 3: Please clarify which groups are being compared. I am assuming those that received the intervention vs not. But then its confusing when we are talking about those that got Pain 101 vs not.
To make Table 5 -9 easier to read-perhaps report the change scores between pre and post intake vs each mean score separately? * those whose p levels were statistically significant.
Discussion:
Whether or not led by a physician or a different clinic, I am assuming this is the first study showing that brief intervention of pain neuroscience with intro to CBT significantly improves outcomes in terms of anxiety and acceptance. This can be stated more clearly at the beginning.
Was “more likely to enroll” measured? Was there a difference in the enrollment in Pain 101 between groups? It would be interesting if acceptance of the diagnosis and willingness to engage in life helps people commit and engage in Pain 101. So would be curious if those rates differed by groups even for those that completed vs didn’t complete the 10 session groups/individual session.
Author Response
Please see attached document of response to Reviewers

Reviewer 2 Report
In this study, the research methodology was strictly well described. However, there are some problems to consider, so I suggest it as follows.
< Materials and Methods>
1. Additionally describe IRB number in ethical issues.
2. The starting point for this experiment is 1st December 2010 to 21st August 2013, which is too old for publication. Describe the reason for this.
3. According to a study presented in Table 2, the participants express part of a chronic pain was diverse. Therefore, it should be described that the pain area is irrelevant to the selection criteria of the study participants.
4. The most important factor in this study is the Cognitive Behavioral Therapy (CBT) program. The researchers hypothesized that this CBT intervention therapy would be effective in reducing chronic pain and anxiety in study participants. Although the theoretical basis for CBT on pain management was described in the introduction, scientifically explain the mechanisms of chronic pain and anxiety reduction of your CBT program (Power- Point® presentation) for these study participants.
5. Describe the production process of the CBT program.
6. The subject's safety from yoga and meditation interventions should be described.
<Discussion>
7. According to the research results presented in Table 2, the areas of chronic pain expressed by the participants were diverse. If so, a discussion is needed to support the conclusion that this study intervention is effective even if the pain site is different.
Reviewer 3 Report
Despite the intriguing and compelling topic, the current paper lacks the following elements: a clear rationale explaining the need for the study, clearly defined objectives, background on previously known CBT interventions in chronic pain, what the proposed intervention adds, and how it differs. Furthermore, I believe the authors should explain why CBT should be performed by a physician rather than a CBT-trained psychologist.
Author Response
Please see attached WORD doc

Round 2
Reviewer 3 Report
Why should CBT be performed by physicians rather than by CBT-trained psychologists? This point remains unaswered.
The authors state that the intervention is based on "their knowledge of scientific literature," rather than previously validated protocols. Why? There is a wide range of evidence on CBT for chronic pain in children. This is the main weakness of this study.
Round 3
Reviewer 3 Report
Authors responded satisfactorly.